# An Experimental Study on Milling Titanium Alloy with a Revolving Cycloid Milling Cutter

**Guangyue Wang \*, Xianli Liu, Tao Chen and Weijie Gao**

Key Laboratory of Advanced Manufacturing and Intelligent Technology, Ministry of Education, Harbin University of Science and Technology, Harbin 150080, China; Xianli.liu@hrbust.edu.cn (X.L.); chentao@hrbust.edu.cn (T.C.); 15104503768@163.com (W.G.)
\* Correspondence: wgy@hrbust.edu.cn; Tel.: +86-186-4624-3223



**Featured Application: This research is beneficial to industrial applications. The likely users could be titanium alloy product manufacturers.**

**Abstract:** In this paper, a revolving cycloid milling cutter was designed with a larger effective cutting helix angle and rake angle than a ball end milling cutter of the same diameter. This new type of milling cutter can solve the problems of low machining efficiency, severe tool wear, and low surface quality in titanium alloy processing. A comparison of the cutting performance of titanium alloys processed by the revolving cycloid milling cutter and the ball end milling cutter was carried out to obtain the variation laws of the cutting force and the processing surface quality under different tool wear conditions. The result shows that the wear zone of the revolving cycloid milling cutter is shallow and wide compared to that of the ball end milling cutter. As the wear speeds up, the spoon-shaped wear gathering zone found in the ball end milling cutter does not happen with the revolving cycloid milling cutter. The revolving cycloid milling cutter can significantly lower the axial force, the tangential force, and the ratio of the axial force to the tangential force with a stable cutting process.

**Keywords:** revolving cycloid milling cutter; wear form; wear mechanism; cutting force; titanium alloy

## 1. Introduction

Featuring high specific strength, high creep corrosion resistance, and high wear resistance, titanium alloy has been extensively used in the fields of aviation, aerospace, energy, and biomedical science [1]. Nevertheless, small tool–chip contact area, large unit cutting force, and high cutting temperature might be caused in the cutting process due to the physical properties of titanium alloy, like low thermal conductivity, small elastic modulus, and large friction coefficient. In general, a ball end milling cutter is utilized to produce the surface finish of titanium alloy. In that case, the forming of the helix angle and the rake angle of the end tooth is limited by the tool manufacturing method and fails to achieve the same design parameters as its cylindrical tooth [2–8]. The machining quality and application of titanium alloy parts are severely restricted by the physical characteristics of titanium alloy and the shortcomings of the existing tool.

A variety of research concerning the cutting technology of titanium alloy has been conducted by scholars at home and abroad. Rashid et al. [9] studied the tool wear mechanism when TC4 was cut at a speed of 150 m/min using an uncoated tool made of hard alloy under dry cutting conditions. In the experiment, crater wear, bond wear, and diffusion wear occurred on the rake face of the tool. Moreover, certain plastic deformation could also be observed on the rake face of the tool. Mou et al. [10] put forward a segmented monitoring method based on cutting vibration signals.

Tests showed that the proposed method can achieve a satisfactory detection result with different cutting parameters. By studying the wear pattern of the blade on a coated tool continuously cutting titanium alloy, Koseki et al. [11] determined that the wear pattern of the rake face of the coated blade is broken without plastic deformation. Further, coating damage was also affected by the interface strength between adhesive materials and coatings, as well as the strength of adhesive materials at high temperatures. Daymi et al. [12] studied the influence of workpiece inclination on surface integrity during the machining of titanium alloys. The findings showed that the average compressive stress decreases with increasing angle of the workpiece for avoiding tool center cutting and tool deflection. The best surface roughness was provided when the workpiece inclination was 15°. Tan et al. [13] carried out a test study on titanium alloy milling with respect to the inclination angle of the ball end milling cutter and obtained the influence law of milling method on the cutting force, tool wear, and its surface integrity. By observing from the test, the cutting force generated by upwards milling along the angle, the best surface quality, and the longest service life of the tool milling downwards along the angle could be obtained. Krishnaraj et al. [14] detected that the cutting depth has maximum influence on the cutting force, while the cutting speed has a significant influence on temperature, after conducting a study on a high-speed TC4 cutting test using a carbide-tipped milling cutter. Liang et al. [15] studied the tool wear and corresponding surface morphology in the processing of fine-grained tools machining TC4 at a high speed. First of all, wear on the rake face and the flank face of the tool was studied in detail from the aspects of wear morphology and chemical elements. This indicated that the wear on the rake face is associated with the combined effect of adhesion, diffusion, and wear, while the major wear mechanism of the flank face is adhesive wear. Next, the influence of tool wear on the corresponding surface morphology was evaluated in regard to surface roughness and defects. Chen et al. [16] solved the problem of chip accumulation by proposing a tool featuring coordinated chip flow with variable strengthening edge. According to the result obtained from research on the wear characteristics of the tool with variable strengthening edge, this tool has superior edge retention and tool durability with obvious advantages in the enhancement of chip flow.

To sum up, research concerning the milling of titanium alloy has mainly focused on the cutting mechanism and the quality of the machining surface with the selection of hard alloy ball end milling cutter [17–22]. When machining small-curvature surfaces for aviation structure parts, the cutting efficiency of a ball end milling cutter is low, the tool wear is serious, and the surface quality of the workpiece is poor. In this paper, a revolving cycloid milling cutter is proposed for this processing condition. Compared with the ball end milling cutter, the revolving cycloid milling cutter [23] presents a stable cutting process, a sharper cutting edge, and even wear on the tool for small-curvature surfaces in the aviation structure parts cutting process due to its large helix angle and working rake angle in actual cutting, presenting certain advantages in cutting performance. In order to further study the cutting performance of the revolving cycloid milling cutter, a TC11 cutting experiment was conducted in this paper using the ball end milling cutter and the revolving cycloid milling cutter, targeted at comparing the wear patterns and wear mechanisms of the two tools. The chip morphology and chip deformation of the two tools under the same cutting conditions were compared. Meanwhile, the influence mechanism of tool wear on the cutting force and workpiece surface quality was studied in different tool wear states. The surface quality of the workpiece was evaluated by measuring the surface topography of the workpiece with a white light interferometer. Lastly, the cutting performance of the revolving cycloid milling cutter was evaluated.

## 2. Experimental Design

The revolving cycloid milling cutter is compared with the ball end milling cutter in this paper, so as to study the wear mechanism of the revolving cycloid milling cutter and the influence of tool wear on the quality of the titanium alloy machining surface. The test was performed on a VDL-1000E three-axis CNC milling machine with a rectangular workpiece made of TC11 with dimensions 150 mm × 50 mm × 80 mm. In order to ensure that the processing technology of the experimental tools was the

same, the five-axis grinding path equation of the revolving cycloid milling cutter was derived based on the cutting edge equation and coordinate transformation. Then, the Saacke five-axis tool grinding center was applied in the grinding of the revolving cycloid milling cutter. The ball end milling cutter was ground using NUM software supplied with the Saacke five-axis tool grinding center. A revolving cycloid milling cutter and a ball end milling cutter with consistent geometric parameters were adopted. The related parameters are presented as follows: the diameter was 10 mm with a helix angle of 30°, a rake angle of 6°, and a clearance angle of 8°. As shown in Figure 1, the titanium alloy was clamped at an inclination angle of 30° in order to have down milling and cutting along the angle.

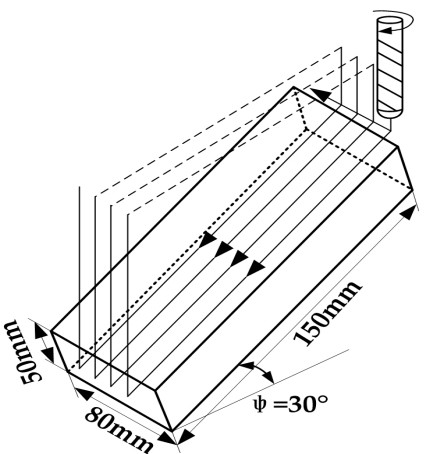

**Figure 1.** Milling mode of the workpiece.

The cutting speed was set to 100 m/min, the feed rate per tooth was 0.06 mm/z, and the axial cutting depth was 0.2 mm, while the cutting width was 0.2 mm. A Kister9171A revolving dynamometer was adopted in the test process to measure the cutting force. When the cutting length was less than 25 m, a VHX-1000 3D microscope system with super depth of field was adopted to observe the worn topography and wearing capacity of the rake face and the flank face of the tool. Meanwhile, the workpiece surfaces obtained at this stage should be maintained. Lastly, a Taylor Surface CCI white light interferometer was adopted to observe the surface morphology of the workpiece. The layout of the experiment site and the tools and workpiece used in the experiment are shown in Figure 2.

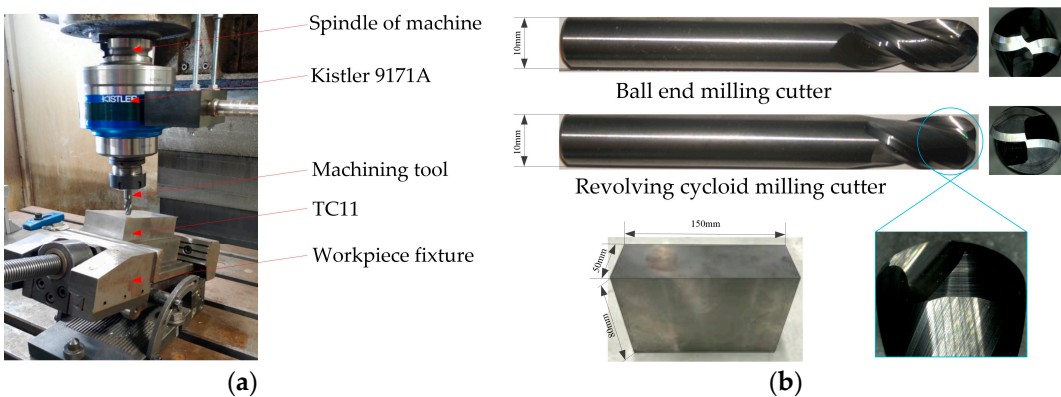

**Figure 2.** Cutting performance test layout: (**a**) Workpiece clamping mode; (**b**) Tools and workpiece.

## 3. Results and Discussion

### 3.1. Structural Characteristics of the Revolving Cycloid Milling Cutter

A revolving cycloid milling cutter was proposed to enhance the quality and efficiency of processing titanium alloy parts. A cycloid rotates around the center of the cutter to form the profile of the cutter end tooth of the revolving cycloid milling cutter. The intersection of the orthogonal helical surface and the tool profile surface is taken as the cutting edge curve. The structural characteristics of the revolving cycloid milling cutter and the ball end milling cutter are compared in Figure 3. The different structural characteristics of the two milling cutters make the difference in the tool angle and tool parameters, which affect the cutting force, tool wear, and surface quality of the workpiece. Therefore, it is necessary to identify the differences between the two tools through numerical simulation before discussing the results.

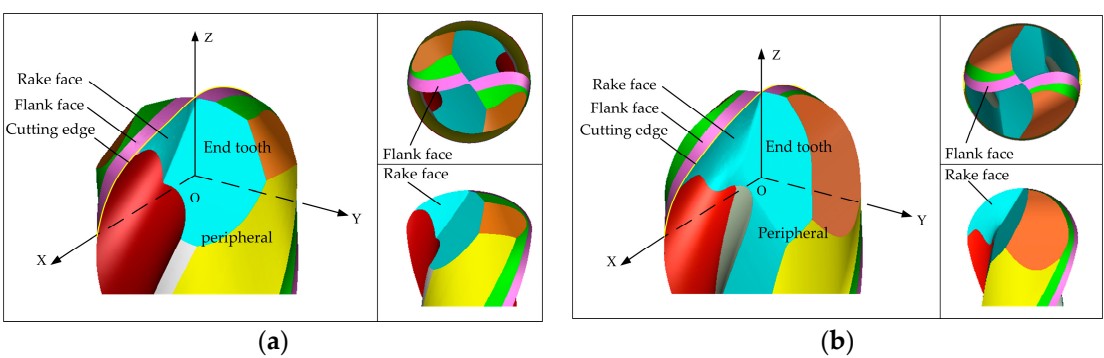

**Figure 3.** Comparison of structural characteristics of the revolving cycloid milling cutter and ball end milling cutter (**a**) Revolving cycloid milling cutter; (**b**) Ball end milling cutter.

It can be seen from the figure that the revolving cycloid milling cutter has short end teeth with a contour similar to a part of an ellipsoid. As a result, the helix angle of the cutting edge generated by the contour of the end tooth is increased rapidly.

Equivalent cutting helix angles of the revolving cycloid milling cutter and the ball end milling cutter with the same radius are compared in Figure 4. Specifically, $\psi$ in the figure represents the machining inclination angle, while point $B_1$ and point $B_2$ are the tool–workpiece contact points of the revolving cycloid milling cutter and the ball end milling cutter, respectively.

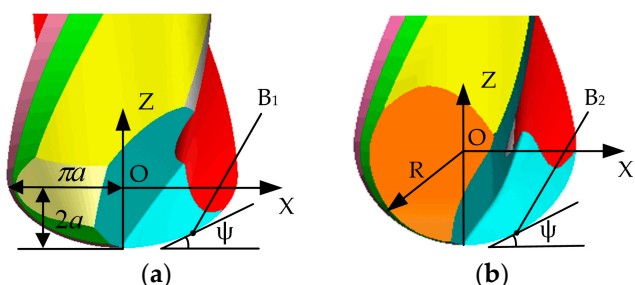

**Figure 4.** Comparison of helix angles: (**a**) Revolving cycloid milling cutter; (**b**) Ball end milling cutter.

$\beta_1$ and $\beta_2$ represent the tool helix angles at point $B_1$ and point $B_2$, respectively. With the machining inclination angle $\psi$ as a parameter, $\beta_1$ and $\beta_2$ were simulated numerically. The result is shown in Figure 5. As can be seen from the figure, $\beta_1$ is greater than $\beta_2$ as the machining inclination angle changes from 0° to 90°. Specifically, when the machining inclination angle is greater than 5° and less than 60°, an apparent advantage can be found in $\beta_1$, indicating that the actual cutting helix angle of the revolving cycloid milling cutter is larger than expected. Furthermore, the angles are interrelated and mutually

affected in the design and cutting process of the milling cutter. According to the literature [24], when the helix angle is smaller than 45° and the cutting depth is less than 0.3 mm, the relationship between the working rake angle $\gamma_{oe}$ of the milling cutter, the helix angle $\beta$, and the normal rake angle $\gamma_n$ can be expressed by Equation (1):

$$\sin \gamma_{oe} = \sin \gamma_n \cos^2 \beta + \sin^2 \beta. \tag{1}$$

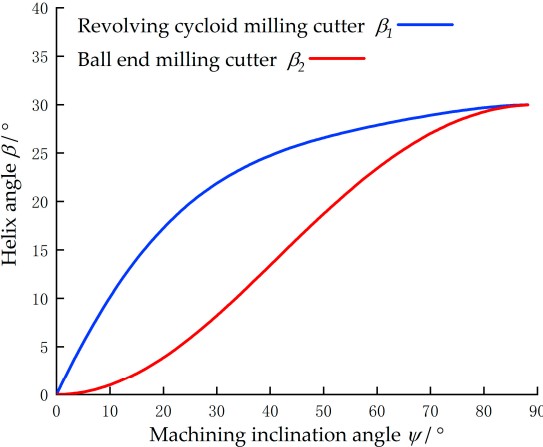

**Figure 5.** Comparison of helix angles under different machining inclination angles.

The equation shows that when the normal rake angle is constant, the working rake angle increases with increasing helical angle. With the helix angle $\beta$ as a parameter, the working rake angle $\gamma_{oe}$ was simulated numerically according to Equation (1). The result is shown in Figure 6.

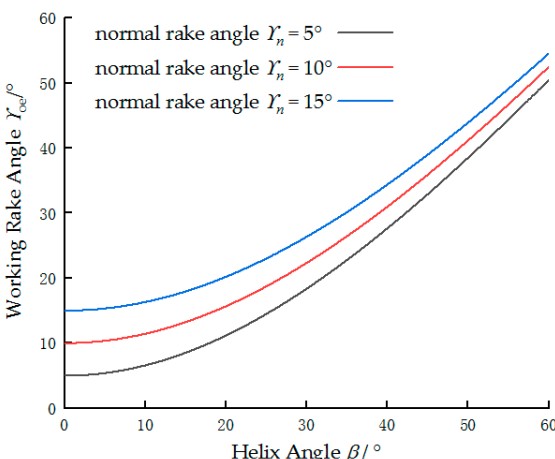

**Figure 6.** Comparison of working rake angles under different helix angles.

It can be observed from the figure that the working rake angle also increases with increasing helix angle. Also, an obvious increasing trend of the working rake angle with the helix angle can be witnessed when the helix angle is greater than 20°.

### 3.2. Tool Wear

After cutting the TC11 using the ball end milling cutter and the revolving cycloid milling cutter under the same milling parameters, the variation laws of the flank with increasing cutting length are presented in Figures 7 and 8, respectively.

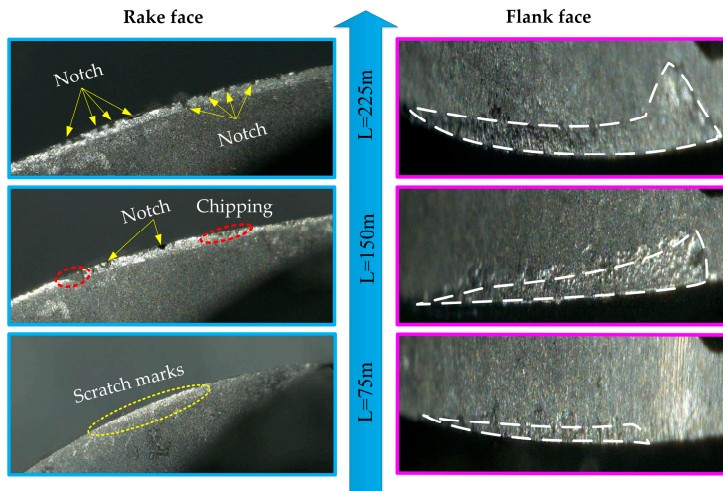

**Figure 7.** The evolution of tool wear in the ball end milling cutter.

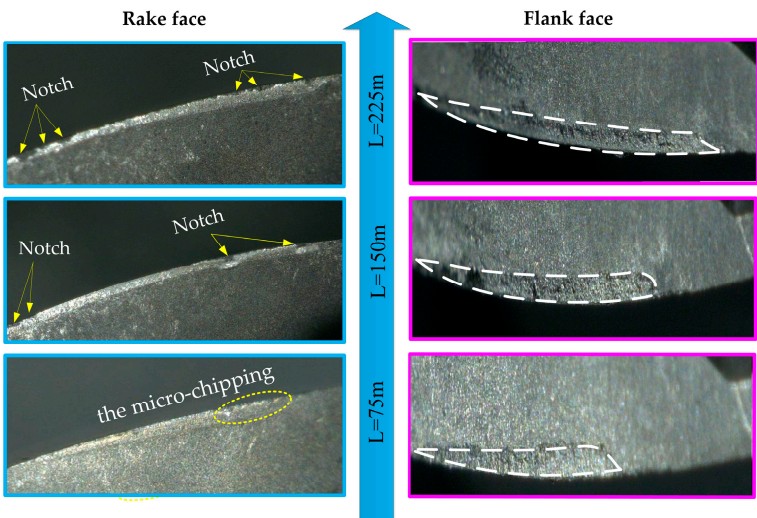

**Figure 8.** The evolution of tool wear in the revolving cycloid milling cutter.

The worn topography of the tool of the ball end milling cutter at different milling lengths under consistent milling parameters is presented in Figure 7. In the figure, when the milling length is 75 m, obvious scrap marks can be seen in the rake face, and a wear belt of a certain width is formed in the flank face. The wear of the tool speeds up with the cutting progress. When the milling length is 150 m, chipping of the tool materials and notches generated under the effect of impact load are observed in the rake face of the ball end milling cutter. Meanwhile, the width of the wear belt in the flank face is extended greatly along the direction away from the cutting edge. When the milling length is 225 m, an increasing number of dense notches can be found in the rake face of the ball end milling cutter, while a spoon-shaped wear belt is formed in the flank face.

The tool wear form of the revolving cycloid milling cutter under comparable cutting conditions is shown in Figure 8. When the milling length is 75 m, slight tool tipping can be observed in the revolving cycloid milling cutter. However, as the cutting process advances, no obvious tool material chipping is observed in the rake face of the revolving cycloid milling cutter, with only a handful of notches when the milling length is 150 m. As the tool wears, there is not such a great number of dense notches as that shown in the ball end milling cutter when the milling length is 225 m. The length of the wear zone on the flank face of the revolving cycloid milling cutter is significantly longer than that of the ball end milling cutter. The wear band changes uniformly with a thin and long morphology.

Considering the cutting method used in the experiment, it can be seen from the figures that the maximum wear zones of the two tools are at the position where the cutting edge of the tool first cuts into the workpiece, rather than at the position of maximum cutting depth. The cutting edge of the revolving cycloid milling cutter is longer and the working rake angle is larger, which reduces the mechanical load during cutting.

### 3.3. Analysis of the Wear Mechanism

The SEM morphology of the flank face with the maximum wear amount when the milling length of the revolving cycloid milling cutter and ball end milling cutter is 225 m is shown in Figure 9.

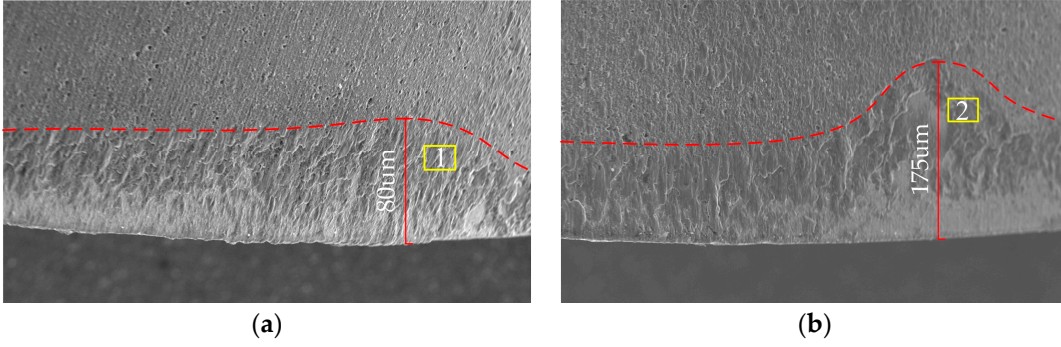

| (a) | (b) |

**Figure 9.** SEM morphology of the maximum wear on the flank face: (**a**) Revolving cycloid milling cutter; (**b**) Ball end milling cutter.

As can be seen from Tables 1 and 2, Ti, O, W, and other elements were found in Zones 1 and 2. Of these, the Ti and O elements come from the workpiece and the external environment. Considering the presence of the above elements and the wear morphology, it was initially determined that bonding wear, diffusive wear, and oxidative wear occur at the wear zone. By further comparison of the element contents at Zones 1 and 2, it was found that contents of the O and Ti elements at Zone 2 were significantly greater than those at Zone 1, indicating that the degree of oxidative wear, bond wear, and diffusion wear at Zone 2 is greater than that at Zone 1. This is mainly because the helix angle and the working rake angle of the cutting edge involved in the cutting are small when the ball end milling cutter cuts, resulting in great stress and high cutting temperature when the tool is cutting. Moreover, the non-cutting stroke temperature is decreased after cut-out of the cutting edge and increases rapidly after cut-in. This reciprocation makes it easier to generate thermal cracks. With cutting progress, spoon-shaped wear is formed finally. The local cutting edge of the tool is damaged by the wear, further increasing the cutting heat and force load. However, with its different structure, the revolving cycloid milling cutter has a larger helix angle and working rake angle at the cutting edge involved in the cutting. Moreover, the cutting edge involving in the cutting is longer, the cutting force is relatively less concentrated, and the thermal load at the maximum cutting depth is relatively small. Oxidative wear, diffusive wear, and thermal cracks appear late with uniform flank wear.

**Table 1.** Energy spectrum analysis (atomic percentage) of Zone 1.

| Tungsten | Titanium | Oxygen | Other Elements |
| --- | --- | --- | --- |
| 11.23% | 8.00% | 14.41% | 66.36% |

**Table 2.** Energy spectrum analysis (atomic percentage) of Zone 2.

| Tungsten | Titanium | Oxygen | Other Elements |
| --- | --- | --- | --- |
| 16.39% | 17.40% | 29.01% | 37.19% |

### 3.4. Cutting Force

A comparison of the cutting force in all directions of the two milling cutters cutting TC11 material at cutting distances of 75 m and 225 m under the same cutting parameters is presented in Figure 10.

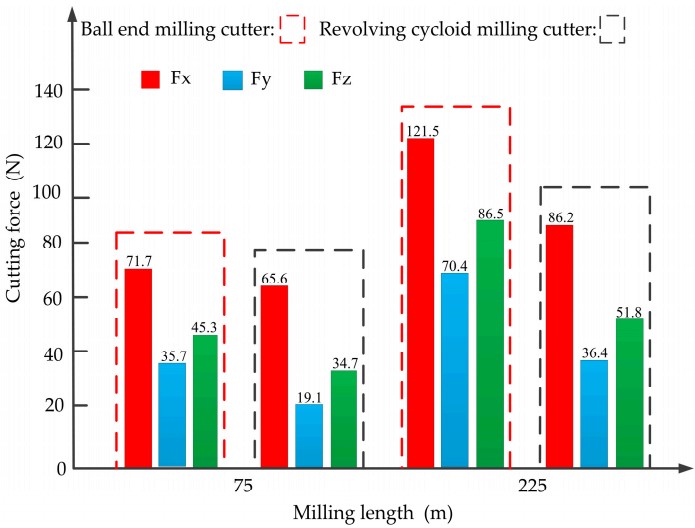

**Figure 10.** Comparison of the maximum cutting force between the two kinds of cutters.

When the milling length is 75 m, the difference between the maximum force values of the two milling cutters in the Fx direction is small, but the maximum force values of the ball end milling cutters in the Fy and Fz directions are significantly larger. When the milling length is 225 m, the tool wear is severe, and the two tools will be changed consistently. Although the variations in cutting forces in all directions of the two tools are consistent, a small increase in the cutting force of the revolving cycloid milling cutter can be observed, mainly because of the effective cutting edge length of the revolving cycloid milling cutter being significantly larger than that of the ball end milling cutter. At the same time, small chip thickness and frictional resistance may be generated in the cutting process. The larger working rake angle also effectively lowers the cutting force, contributing to low wear of the tool. At this time, the tool wear of the ball end milling cutter is more serious than that of the revolving cycloid milling cutter, with a significant increase in cutting forces in all directions.

The ratios of cutting forces under different cutting distances of the two milling cutters are compared in Figure 11. It can be seen from the figure that the variation magnitudes of the tangential force and the axial force of the two milling cutters are remarkably greater than that of the radial force. The ratios of axial force to tangential force are 0.63, 0.53 and 0.71, 0.60, respectively, at milling lengths of 75 m and 225 m. To be specific, the ratio of axial force to tangential force of the revolving cycloid milling cutter is small. With cutting progress, the ratios of axial force to tangential force of the two milling cutters increased. The increase range of the revolving cycloid milling cutter is small, which indicates that the cutting process is stable and the tool wear is small. Further, it can maintain good cutting conditions even if the tool is worn.

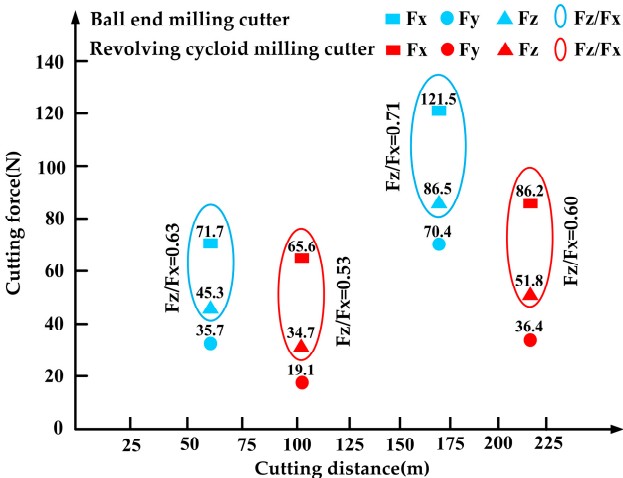

**Figure 11.** Cutting force ratios of the two milling cutters at different milling distances.

### 3.5. Chip Morphology

The undeformed chip morphology of the two milling cutters is compared in Figure 12. Both of these chips are theoretical undeformed chip models made by using a Boolean operation module in NX10.0 software. As there is a small part involved in cutting in the actual cutting process of the tool, the difference between the revolving cycloid milling cutter and the ball end milling cutter in contour cannot be presented in the chip morphology. Therefore, the two chip shapes are similar. The chip of the revolving cycloid milling cutter is slightly larger, while the thickness is slightly less at the thickest part of the chip.

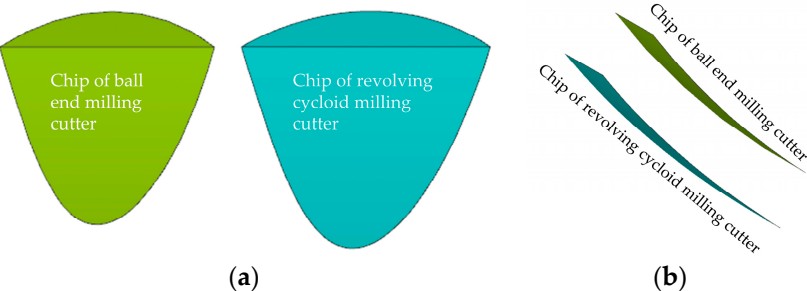

(**a**)  (**b**)

**Figure 12.** Comparison of undeformed chip morphology of the two milling cutters: (**a**) Front view; (**b**) Side view.

The chip deformation of the two milling cutters is compared in Figure 13. It can be seen that chips from the two milling cutters are deformed to different degrees, where the deformation of chips produced by the revolving cycloid milling cutter is slight. This is mainly because the larger working rake angle of the revolving cycloid milling cutter can strengthen the shearing effect of the tool on the processed material and effectively decrease the deformation and the internal stress of the workpiece material to be machined.

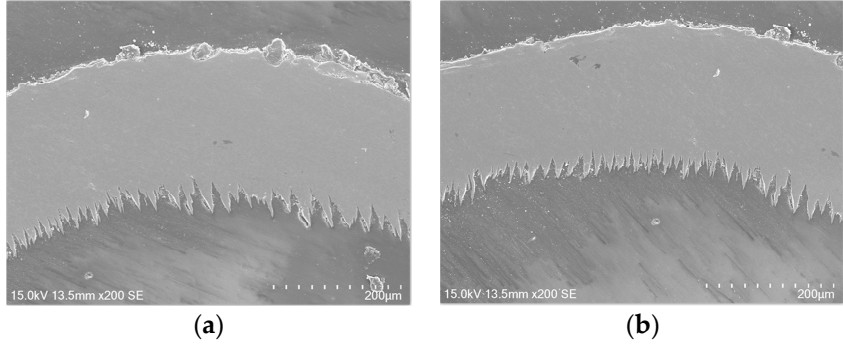

**Figure 13.** The chip micromorphology of the two milling cutters: (**a**) Revolving cycloid milling cutter; (**b**) Ball end milling cutter.

### 3.6. Machining Surface Topography

Figure 14 shows the surface topography of the workpiece with a milling length of 225 m for the two cutters. When the milling length is 225 m, the severely worn cutting edge intensifies the friction and extrusion of the flank face due to the small helix angle and working rake angle of the ball end milling cutter in the cutting process. As a result, built-up edges might be generated from workpiece materials being bonded on the rake face, which might lead to defects such as tears and pits on the workpiece surface. Meanwhile, a spoon-shaped wear zone is formed in the flank face, resulting in a number of material residues on the workpiece surface. Consequently, a dramatic drop in quality of the machining surface can be witnessed after the occurrence of the above phenomenon. Nevertheless, the machining surface quality of the revolving cycloid milling cutter is more ideal, since it has a large equivalent cutting helix angle and working rake angle, as well as a long effective cutting edge. Moreover, with slight wear in the cutter, the rake face, the cutting edge of the cutter, and the flank face can still be operated stably.

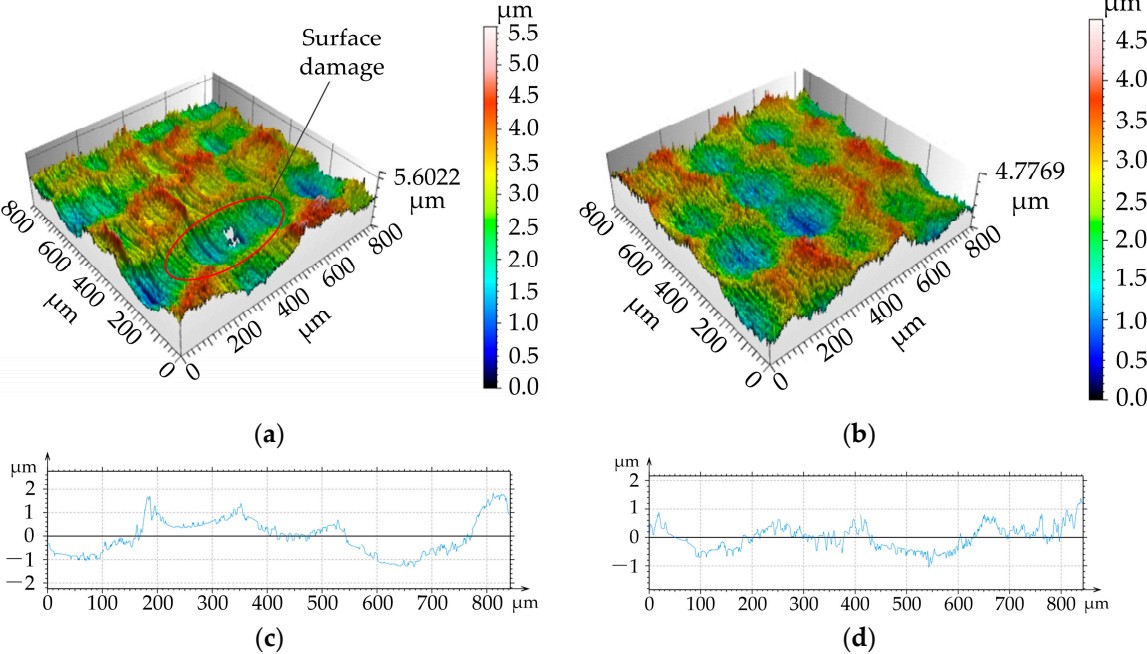

**Figure 14.** Surface topography of the workpiece when the milling length is 225 m: (**a**) 3D machining surface topography (ball end milling cutter); (**b**) 3D machining surface topography (revolving cycloid milling cutter); (**c**) 2D contour (ball end milling cutter); (**d**) 2D contour (revolving cycloid milling cutter).

## 4. Conclusions

1.  Through innovative design of the cutter profile, a revolving cycloid milling cutter was proposed. This kind of milling cutter design can improve the helical angle and the working rake angle of the cutter. By means of numerical simulation and comparison, we showed that the helical angle and working rake angle of the revolving cycloid milling cutter are at least 31% and 22% higher than those of the ball end milling cutter.

2.  The cutting performance of TC11 with the revolving cycloid milling cutter and ball end milling cutter was compared. The result shows that the wear zone of the revolving cycloid milling cutter is shallow and wide, compared to that of the ball end milling cutter. As the wear speeds up, the spoon-shaped wear gathering zone found with the ball end milling cutter does not occur with the revolving cycloid milling cutter.

3.  The revolving cycloid milling cutter can significantly lower the axial force, the tangential force, and the ratio of axial force to tangential force with stable force variation in the cutting-in and cutting-out process. In the case of tool wear, the cutting forces in the Fx, Fy, and Fz directions increased by 24%, 48%, and 33%, respectively. The ratio of axial force to tangential force only increased from 0.53 to 0.6, showing good cutting stability.

4.  The chip generated by the revolving cycloid milling cutter after cutting TC11 is slightly larger than that by the ball end milling cutter, whereas it has a smaller degree of chip deformation. No defect occurring in machining by the ball end milling cutter was found on the TC11 surface by the revolving cycloid milling cutter. In comparison with that of the ball end milling cutter, the cutting performance of the revolving cycloid milling cutter is outstanding.

**Author Contributions:** Conceptualization, G.W. and T.C.; methodology, X.L.; software, W.G.; validation, W.G. and G.W.; formal analysis, T.C.; investigation, W.G.; writing—original draft preparation, G.W.; writing—review and editing, W.G.; visualization, G.W.; supervision, X.L.; project administration, T.C.; funding acquisition, X.L. All authors have read and agree to the published version of the manuscript.

**Funding:** This research was funded by the National Natural Science Foundation's international (regional) cooperation and exchange project: The basis and application of intelligent cutting technology based on open CNC system, grant number 51720105009.

**Conflicts of Interest:** The authors declare no conflict of interest.

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
