# Peer review of "An Experimental Study on Milling Titanium Alloy with a Revolving Cycloid Milling Cutter"

_applsci, doi:10.3390/app10041423_

Round 1

Reviewer 1 Report

Once I have verified that all the changes requested by me and by the editor have been taken into account, I consider that the paper can be published in  present form

Author Response

Once I have verified that all the changes requested by me and by the editor have been taken into account, I consider that the paper can be published in  present form

Response: The necessary changes have been made.

The authors thank you very much for your guidance and help.

Reviewer 2 Report

The resubmitted paper includes most corrections indicated by the reviewer. The English language of the paper and the References are also improved. However the Figure 13 - Surface topography of the workpiece (formerly Fig. 14) is still not visible. In opinion of the reviewer the overall quality of reviewed manuscript is now on high level. Because of this, I recommend this paper for publication in Applied Sciences journal.

Author Response

The resubmitted paper includes most corrections indicated by the reviewer. The English language of the paper and the References are also improved. However the Figure 13 - Surface topography of the workpiece (formerly Fig. 14) is still not visible. In opinion of the reviewer the overall quality of reviewed manuscript is now on high level. Because of this, I recommend this paper for publication in Applied Sciences journal.

Response: The figure has been modified to make the results clearer.

The authors thank you very much for your guidance and help.

Reviewer 3 Report

Paper was corrected and can be accepted "as-is" in the present state.

Author Response

Paper was corrected and can be accepted "as-is" in the present state.

Response: The authors thank you very much for your guidance and help.

Reviewer 4 Report

Some comments:

- The summary is too long. It has to be more precise with the information from the research - no introduction is needed (lines 12-16, 27-30)

- Line 93 – how was the monitoring of surface quality studied??? By which technique?

- Part 2 (line 95) - it must be moved to part 4. Results and Discussion

- A clearer presentation of Part 3 - Experimental Design / add Methods of characterisation - too complicated, no photos - picture 5a

- Part 4 – requires a better interpretation by combination with Part 2 from line 95

- Presentation of wear mechanism (graphical) is needed, Figure 8 – to small – no visible scale line

- The conclusion needs to be remodelled, because it is almost identical to the summary (specifically, write down what the results are) and what readers may see as novelty.

Author Response

Some comments:

- The summary is too long. It has to be more precise with the information from the research - no introduction is needed (lines 12-16, 27-30)

Response: The lines 12-16, 27-30 have been deleted in order to clearly represent the information of the research.

- Line 93 – how was the monitoring of surface quality studied??? By which technique?

Response: The surface quality of the workpiece is evaluated by measuring the surface topography of the workpiece with a white light interferometer. At present, residual stress has not been measured. Evaluation methods for workpiece surface quality have been added to the paper, and marked in red.

- Part 2 (line 95) - it must be moved to part 4. Results and Discussion

Response: Part 2 has been moved to Part 4 and the title is marked red.

- A clearer presentation of Part 3 - Experimental Design / add Methods of characterisation - too complicated, no photos - picture 5a

Response: A figure has been added to illustrate the clamping and milling mode of the workpiece. The name of the figure is marked in red.

- Part 4 – requires a better interpretation by combination with Part 2 from line 95

Response: For better explanation, some content has been added in combination with the content of Part 2, and marked in red.

- Presentation of wear mechanism (graphical) is needed, Figure 8 – too small – no visible scale line

Response: The key parts of figure 8 have been enlarged and dimensioned at the location of maximum wear. The name of the figure has been marked in red.

- The conclusion needs to be remodelled, because it is almost identical to the summary (specifically, write down what the results are) and what readers may see as novelty.

Response: The conclusion has been revised to highlight the results and innovations of the study. The changes have been marked in red.

The authors thank you very much for your professional advice.

Round 2

Reviewer 4 Report

/